# Immunotherapy and Hypofractionated Radiotherapy in Older Patients with Locally Advanced Cutaneous Squamous-Cell Carcinoma of the Head and Neck: A Proposed Paradigm by the International Geriatric Radiotherapy Group

**DOI:** 10.3390/cancers15204981

**Published:** 2023-10-13

**Authors:** Nam P. Nguyen, Juliette Thariat, Olena Gorobets, Vincent Vinh-Hung, Lyndon Kim, Sergio Calleja Blanco, Maria Vasileiou, Meritxell Arenas, Thandeka Mazibuko, Huan Giap, Felix Vincent, Alexander Chi, Gokoulakrichenane Loganadane, Mohammad Mohammadianpanah, Agata Rembielak, Ulf Karlsson, Ahmed Ali, Satya Bose, Brandi R. Page

**Affiliations:** 1Department of Radiation Oncology, Howard University, Washington, DC 20059, USA; satya.bose@howard.edu; 2Department of Radiation Oncology, Francois Baclesse Cancer Center, 14000 Cain, France; jthariat@gmail.com; 3Department of Oral Surgery, University of Martinique, 97213 Martinique, France; ellen.gorobets@igrg.ch; 4Department of Radiation Oncology, Institut Bergonie, 33076 Bordeaux, France; anhxang@gmail.com; 5Division of Neuro-Oncology, Mount Sinai Hospital, New York, NY 10029, USA; lyndon.kim@mssm.edu; 6Department of Oral and Maxillofacial Surgery, Howard University, Washington, DC 20059, USA; seca@umich.edu; 7Department of Pharmacy, School of Health Sciences, National and Kapodistrian University of Athens, 15771 Athens, Greece; 8Department of Radiation Oncology, Sant Joan de Reus University Hospital, University of Rovira, I Virgili, 43204 Tarragona, Spain; meritxell.arenas@gmail.com; 9Department of Radiation Oncology, International Geriatric Radiotherapy Group, Washington, DC 20001, USA; firstblackoncologist@gmail.com (T.M.); karlssonulf30@gmail.com (U.K.); 10Department of Radiation Oncology, Medical University of South Carolina, Charleston, SC 29425, USA; hbgiap@gmail.com; 11Department of Surgery, Southern Regional Health System, Lawrenceburg, TN 29425, USA; fbvincent@gmail.com; 12Department of Radiation Oncology, Xuanwu Hospital, Capital Medical University, Beijing 101125, China; achiaz2010@gmail.com; 13Department of Radiation Oncology, CHU Mondor, 94000 Creteil, France; gloganadane@yahoo.com; 14Colorectal Research Center, Department of Radiation Oncology, Shiraz University of Medical Sciences, Shiraz 71348-14336, Iran; mohpanah@gmail.com; 15Department of Radiation Oncology, The Christie NHS Foundation Trust, Manchester M20 4BX, UK; agata.rembielak@nhs.net; 16Division of Cancer Sciences, Faculty of Biomedicine and Health, School of Medical Sciences, The University of Manchester, Manchester M13 9PL, UK; 17Division of Hematology Oncology, Howard University, Washington, DC 20059, USA; ahali@huhosp.org; 18Department of Radiation Oncology, Johns Hopkins University, Baltimore 21218, MD, USA; brandi.page@gmail.com

**Keywords:** older, cutaneous squamous cell, locally advanced, CPIs, hypofractionated radiotherapy

## Abstract

**Simple Summary:**

Older patients are at risk of skin cancer. Head and neck cutaneous squamous-cell carcinoma is due to prolonged sun exposure. When the disease is advanced, it requires a combination of an operation to remove the cancer, followed by radiotherapy for possible cure. However, older patients may not tolerate surgery. Recent studies show that this type of cancer may be very vulnerable to immunotherapy. We propose to combine immunotherapy with a short treatment course of radiotherapy instead of the conventional surgery and radiotherapy because those two treatments may work together to improve cure rates. However, clinical studies should be performed to verify our hypothesis.

**Abstract:**

Cutaneous skin carcinoma is a disease of older patients. The prevalence of cutaneous squamous-cell carcinoma (cSCC) increases with age. The head and neck region is a frequent place of occurrence due to exposure to ultraviolet light. Surgical resection with adjuvant radiotherapy is frequently advocated for locally advanced disease to decrease the risk of loco-regional recurrence. However, older cancer patients may not be candidates for surgery due to frailty and/or increased risk of complications. Radiotherapy is usually advocated for unresectable patients. Compared to basal-cell carcinoma, locally advanced cSCC tends to recur locally and/or can metastasize, especially in patients with high-risk features such as poorly differentiated histology and perineural invasion. Thus, a new algorithm needs to be developed for older patients with locally advanced head and neck cutaneous squamous-cell carcinoma to improve their survival and conserve their quality of life. Recently, immunotherapy with checkpoint inhibitors (CPIs) has attracted much attention due to the high prevalence of program death ligand 1 (PD-L1) in cSCC. A high response rate was observed following CPI administration with acceptable toxicity. Those with residual disease may be treated with hypofractionated radiotherapy to minimize the risk of recurrence, as radiotherapy may enhance the effect of immunotherapy. We propose a protocol combining CPIs and hypofractionated radiotherapy for older patients with locally advanced cutaneous head and neck cancer who are not candidates for surgery. Prospective studies should be performed to verify this hypothesis.

## 1. Introduction

Cutaneous cancer is a disease of the elderly. Prolonged skin exposure to ultraviolet light leads to chronic skin damage and cancerous transformation in exposed areas such as the head and neck [1,2]. Among non-melanoma skin cancer, basal-cell and squamous-cell carcinoma are prevalent. However, compared to basal-cell cancer, the squamous histology portends a worse prognosis with a higher risk of loco-regional recurrence and in some patients with high-risk features such as poorly differentiated histology and peri-neural invasion, the potential for nodal and distant metastases [3,4,5]. For patients with locally advanced disease, surgical resection followed by postoperative irradiation are frequently recommended [6]. However, older cancer patients may not be candidates for surgery due to multiple comorbidities and frailty [7]. In addition, surgical resection may lead to a higher risk of postoperative complications, impaired functional and cosmesis outcomes, and a reduced quality of life (QOL) in frail patients [8,9]. In patients who are inoperable, radiotherapy alone or combined with systemic therapy are offered as alternatives [10,11,12]. Although age should not be the sole deciding factor in the management of older adults with cSCC, treatment burden, especially in complex and extensive treatments, should be carefully considered against patient’s fitness and wishes [13]. In contrast to basal-cell carcinoma, radiotherapy alone for cutaneous squamous-cell carcinoma (cSCC) with an advanced stage has been reported with a high rate of local–regional failure and poor survival [10]. Chemotherapy alone or combined with radiotherapy has been proposed to improve the prognosis [11,12]. However, in frail patients, chemotherapy may lead to excessive mortality regardless of age [14]. The replacement of chemotherapy with cetuximab in combination with radiotherapy was also associated with significant toxicity in older and frail patients requiring hospital admission [15].

Hence, there is a pressing need for a novel treatment algorithm tailored specifically to older patients with locally advanced head-and-neck cSCC, aiming to enhance their survival rates while maintaining their quality of life. Recently, immunotherapy with check-point inhibitors (CPIs) has emerged as a promising agent for cSCC because of its efficacy and safety profile [16,17,18]. Older patients with cancer and minorities are frequently excluded from clinical trials [19,20,21]. Up to 67% of the trials listed in ClinicalTrials.gov have upper age limits and/or exclusion criteria that limit the recruitment of older patients [19]. A similar analysis of 1012 clinical trials registered on ClinicalTrials.gov also corroborates that not only are older cancer patients excluded from clinical trials, but minorities and women are also less likely to be enrolled [21]. As an international organization devoted to the care of older cancer patients, the International Geriatric Radiotherapy Group (http://www.igrg.org (accessed on 8 October 2023)) would like to investigate if immunotherapy combined with radiotherapy could be an option to improve survival and QOL for those vulnerable patients [22,23,24].

## 2. Prevalence of Program Death Ligand 1 (PD-L1) Receptors in Patients with cSCC

Chronic exposure to the sun produces ultraviolet DNA damage, a high mutation burden, and an increase in PD-L1 expression in cSCC [25]. Locally advanced cSCC frequently expresses a higher PD-L1 expression compared to a disease of an early stage. Those tumors are often large, poorly differentiated, with perineural and regional lymph nodes invasion, features which are associated with a high PD-L1-level expression [26,27,28,29,30,31]. Thus, it is not surprising that immunotherapy with checkpoint inhibitors (CPIs) produces an excellent and long-lasting response in selected patients with locally advanced and metastatic disease. Other biomarkers such as tumor-infiltrating lymphocyte (TIL) and tumor mutational burden (TMB) can also be tested to assess tumor response to immunotherapy, and could be included with PD-L1 testing in next-generation sequencing (NGS) to assess tumor response to other agents in most solid tumors [32,33]. As our knowledge about tumor molecular biology is expanding, NGS may improve the therapeutic options for metastatic cutaneous-cell carcinoma and may guide therapy in the case of tumor refractory to standard treatment [34,35]. However, given the cost and time delay in obtaining the results, it is doubtful that NGS will be tested in the foreseeable future for cSCC.

## 3. Effectiveness of CPIs in the Treatment of cSCC

In clinical trials, two PD-1 antibody agents, cemiplimab and pembrolizumab, and two PD-L1 antibodies, nivolumab and cosibelumab, have proven to be effective for cSCC [16,18,36].

In the neoadjuvant setting, cemiplimab was investigated in 79 patients of stages II, III, and IV (M0), who underwent surgical resection with curative intent after up to four doses of 350 mg every three weeks [36]. A complete pathological response was reported among 40 patients (51%). Fourteen patients (18%) developed grade 3 or higher side effects. No death was reported. However, 11 patients (13.9%) had disease progression during the drug administration. The PD-L1 status of those with disease progression was unclear as only 56 patients in the study had PD-L1 status assessed before surgery. Those with positive PD-L1 (>1%) had a better clinical response. The complete pathological response was 20% and 54% for patients with PD-L1 expression <1% and 1% or more, respectively. Corresponding clinical response was 47% and 76%, respectively. The study corroborated the efficacy of cemiplimab in the neoadjuvant setting in a previous pilot study [18] but highlighted the importance of having the biomarker at baseline.

Among patients with locally advanced cSCC who were unresectable, 78 received cemiplimab 3 mg/kg every two weeks for up to 96 weeks. At a median follow-up of 9 months, 34 patients (44%) achieved a response, 13% complete, and 31% partial. Grade 3–4 side effects occurred in 34 patients (44%). One died from aspiration pneumonia. As PD-L1 status was not assessed, it is unclear whether there was a correlation between clinical response and PD-L1 expression [37]. Patients who achieved a clinical response experienced an excellent QOL [16]. Thus, although encouraging, these preliminary studies emphasized the need to incorporate PD-L1 or other biomarkers for immunotherapy into the study protocol to better assess response to treatment and to select patients more likely to benefit from the treatment.

In the real-world setting, cemiplimab has been reported to have higher grade 3–4 side effects, which likely reflected the patients’ poor performance statuses compared to the ones recruited in clinical trials [38]: 45% experienced severe side effects, requiring treatment discontinuation in nine (41%). One patient died (4.5%). However, 47% experienced disease control similar to the phase II study.

Pembrolizumab has also been reported to be effective for unresectable and metastatic cSCC. A total of 57 patients were treated with 200 mg of pembrolizumab every three weeks until disease progression or unacceptable toxicity [39]. Response rate was 42%, 7%, and 35% for total response, complete response (CR), and partial response, respectively. A total of 34 patients (60%) achieved disease control. The response rate was significantly better for patients with positive PD-L1 defined as 1% or above, and was 55% and 17% for positive and negative PD-L1, respectively. Grade 3 or higher toxicity occurred in 7%. No death was reported. The study was updated when a total of 159 patients were recruited [40]. Overall response, CR, and PR were 40.3%, 12.6%, and 27.7%, respectively. A total of 56% achieved disease control. Grade 3–5 side effects occurred in 11.9%. Two patients died (1.3%).

Preliminary results suggest that nivolumab at a dose of 3 mg/kg every two weeks until disease progression, unacceptable toxicity, or 12 months of treatment may achieve a good response rate in cSCC [41]. Response rate was 58.3% among 24 enrolled cSCC patients, but there was no complete response. Grade 3–4 toxicity occurred in six patients (25%), requiring treatment discontinuation in one. Another PD-L1 agent, cosibelumab, achieved a response rate of 47% in 103 patients with metastatic cSCC [42]. Grade 3–4 toxicity was 36%. In the metastatic setting, the optimal duration of CPIs remains unknown. Unlike chemotherapy agents, immunotherapy has the potential to achieve long-term remission. However, its benefit has to be weighted against long-term side effects with cumulative dose, especially among older patients [43]. Thus, most protocols would specify a pre-determined length of treatment, which would be discontinued if severe side effects developed or disease progression occurred.

Among older patients with advanced and recurrent cSCC, immunotherapy with CPIs is also effective with acceptable toxicity. Samaran et al. [44] reported 63 cSCC patients with a minimum age of 70 years (median 82) who received various CPIs for their cSCC after the failure of their previous therapy. The response rate was 57.1%, 19%, and 38.1% for total response, CR, and PR, respectively. Grade 3 toxicity was 28.5%. Three patients died (4.7%) from tumor hemorrhage (*n* = 2) and pneumonia (*n* = 1). Strippoli et al. [45] also corroborated the high response rate of CPIs (76%) among 30 older and frail patients with advanced or metastatic cSCC. Thus, CPIs with either anti PD-1 or anti PD-L1 agents are effective for cutaneous squamous-cell carcinoma. It is unclear which agent carries less toxicity. Table 1 summarizes response rate and toxicity of CPIs in cSCC.

In summary, immunotherapy with CPIs was effective in phase II studies of cSCC with acceptable toxicity. About half of the patients achieved a response rate, which may be dependent on their PD-L1 status. Both anti-PD-1 or anti-PD-L1 agents are effective for cutaneous squamous-cell carcinoma. It remains unclear which ones carry less toxicity. Could the addition of radiotherapy to immunotherapy improve this clinical response to achieve a better QOL for older patients with locally advanced cSCC?

## 4. Potential of Radiotherapy as a Synergistic Modality to Improve Response Rate to Immunotherapy in Patients with cSCC

Even though PD-L1 is not a perfect biomarker to predict tumor response to CPIs, high PD-L1 expression defined as 50% or more or a high tumor proportion score (TPS) often indicates a high response to immunotherapy regardless of tumor histology. As an illustration, among 11 patients with recurrent and metastatic cSCC who developed excellent response to CPIs, those who achieved CR had a PD-L1 expression of at least 30% [46]. Another method, the combined positive score (CPS), is also effective and similar to TPS to predict tumor response in head and neck cancer [47]. Excellent survival was reported with less toxicity among patients with a high TPS score [48,49]. Could radiotherapy increase the PD-L1 expression of tumor cells, thus enhancing the tumoricidal effect of immunotherapy, or alternatively, turning a tumor devoid of PD-L1 to a positive one?

Radiotherapy produces inflammation in the tumor environment through DNA breakage. Through a complex mechanism which involves a DNA damage signaling pathway, interferon γ (IFNγ) signaling, the cyclic GMP–AMP synthase–stimulator of interferon genes (cGas–STING) pathway, and the epidermal growth factor receptor (EGFR) pathway, the PD-L1 receptor of the tumor cells is upregulated [50]. As a result, following radiotherapy, PD-L1 expression increases significantly in proportion to the dose of radiation delivered both in the in vitro and in vivo setting [51,52]. The increase in PD-L1 expression following irradiation reflects a protective mechanism of the tumor to escape cell death from infiltrating T cells, which are attracted into the tumor microenvironment by radiation-induced inflammation. Thus, it could be used as a clinical strategy to enhance response to immunotherapy [53].

Clinical studies support the upregulation of PD-L1 receptor by radiotherapy. In patients with cervical cancer who had biopsy twice before radiotherapy and after 1200 cGy in four fractions, a significant elevation of PD-L1 expression was reported. The prevalence of PD-L1 was 45% and 87% before and after 1200 cGy, respectively [54]. Thus, an increased level of PD-L1 occurred rapidly after irradiation. Another study corroborated the positive correlation between radiotherapy and PD-L1. Among 75 patients who had locally advanced squamous-cell cervical carcinoma and were treated with radiotherapy or chemoradiation, only 5% was positive for PD-L1 in the pre-treatment biopsy. Following 1000 cGy in five fractions, a repeated cervical biopsy demonstrated a 54% PD-L1 positivity [55]. The potential of radiotherapy to convert a PD-L1 negative tumor to a positive one was illustrated by Patel et al. [56]. Among 46 patients who had stages II and III sarcoma requiring preoperative irradiation, no patient had PD-L1 expression before surgery. After a dose of 5000 to 5040 cGy in 25 fractions, five patients (10.9%) became PD-L1 positive. Interestingly, the percentage of tumor-associated macrophages (TAM) expressing PD-L1 or M2 also increased from 15.2% to 45.7% after irradiation. As TAM role is to suppress CD8+ T cell infiltration through cytokines secretion, this may indicate another strategy for tumor cells to evade killing by the immune system through producing an immunosuppressive environment [57]. The increase in PD-L1 expression following irradiation was also observed in lung cancer and rectal cancer [58,59]. The elevation of PD-L1 expression is specific to radiotherapy. In NSCLC undergoing chemoradiation or systemic therapy with chemotherapy or tyrosine kinase inhibitor, a significant increase in PD-L1 expression was observed after chemoradiation but not following drug therapy [58].

The ability of tumor cells to evade the radiotherapy-induced immune response, which in turn make them more vulnerable to CPIs, has been investigated in the clinic. Narits et al. [60] reported the immune response of a patient with a metastatic lung adenocarcinoma, which was PD-L1 negative on initial biopsy. The patient developed progressive disease despite chemotherapy, which required the hypofractionated radiotherapy of 4500 cGy in 15 fractions to the mediastinum. Following initial stabilization and despite chemotherapy, the mediastinal mass grew in size again. A repeat biopsy was performed and demonstrated positive-PD-L1 tumor cells with a TPS score of 100%. Pembrolizumab was initiated and produced a partial response of all the lesions. The patient remained in remission after the discontinuation of immunotherapy, was able to work full-time for over two years, and was still in remission after his last follow-up visit. Thus, hypofractionated radiotherapy in this case may provide long-term remission with immunotherapy through the upregulation of PD-L1 receptors. Even though the optimal sequencing of radiotherapy and immunotherapy remains unknown with conflicting preclinical data and needs to be determined through clinical trials, this case report suggests that patients who initially lack PD-L1 expression in the tumor may benefit from radiotherapy first to induce PD-L1 expression in the tumor followed by immunotherapy for a better response. Patients with high PD-L1 expression in the tumor may benefit from upfront immunotherapy as they likely will have a good response to the treatment. Preliminary studies in patients undergoing pembrolizumab alone or combined with chemotherapy with in head-and-neck-cancer patients with a CPS > 1 suggested a superior survival compared to cetuximab and chemotherapy [61]. This CPS selection is critical as pembrolizumab combined with radiotherapy failed to improve survival among head-and-neck-cancer patients unfit for cisplatin compared to cetuximab and radiotherapy [62]. However, grade 3–4 toxicity was significantly lower in the pembrolizumab group and illustrated the safety of this regimen.

Another potential effect of high-dose irradiation is the modulation of the immune response in the body causing the shrinkage of other metastatic sites outside of the radiation field, described as the abscopal effect, which in rare cases, may improve survival. The molecular mechanism of the abscopal effect is complex, but is thought to be mediated through CD8+T cells [63]. There has been a case report of a patient with multiple synchronous cSCC who underwent brachytherapy for one lesion and developed a spontaneous regression of the non-irradiated ones [64]. Thus, adding immunotherapy to irradiation may produce a better response due to their synergy.

## 5. Tolerance of Older Patients with Cutaneous Squamous-Cell Carcinoma to Hypofractionated Radiotherapy

Hypofractionated radiotherapy is a convenient treatment schedule for older cancer patients due to their limited mobility and difficulty with transportation if they live a long distance from the cancer center [65,66]. Hypofractionated radiotherapy was well tolerated among older patients with cSCC who were not candidates for surgery [67,68,69,70,71,72]. A weekly schedule was proposed to minimize transportation issues. Among 18 patients aged 75 years or above (median: 89 years old) who received 800 cGy every week for seven to eight weeks for locally advanced cSCC, bleeding and pain resolved in all patients with no grade 3–4 toxicity [67]. The efficacy of the once or every other week fractionation was corroborated in another study of frail and old patients with cSCC [68,69]. Many dose fractionations were proposed to achieve a biology-equivalent dose (BED) up to 10,000 cGy, which may provide a reasonable chance for local control with good cosmesis if the tumor is localized in the head and neck area [70]. A systematic review of 40 studies of patients undergoing hypofractionated radiotherapy for non-melanoma skin cancer reported excellent local control despite the heterogeneity of the patient population [73]. A similar systemic review of older head-and-neck-cancer patients undergoing hypofractionated radiotherapy demonstrated that this radiotherapy technique is well tolerated in this patient population with good local control [74]. In the era of the COVID-19 pandemic, hypofractionated radiotherapy also limits the exposure of this vulnerable population to the virus [75]. Thus, any treatment schedules ranging from daily to one or bi-weekly could be tailored to patients’ needs depending on the patient’s frailty status, mobility, and home distance from the cancer center. Flexibility should be the key in designing treatment schedules for older patients to improve their QOL.

## 6. Feasibility of Hypofractionated Radiotherapy and Immunotherapy for Older Patients with cSCC

The feasibility of radiotherapy concurrently with immunotherapy was investigated in four patients with inoperable locally advanced cSCC. An amount of 2 mg/kg of Pembrolizumab every three weeks was administered concurrently with hypofractionated radiotherapy [76]. Two patients achieved long-lasting CR. There was no grade 3–4 toxicity. Another study corroborated the safety profile of concurrent immunotherapy and radiotherapy: 12 patients with locally advanced (*n* = 1) or metastatic disease (*n*= 11) received 350 mg of cemiplimab every three weeks with hypofractionated radiotherapy to a BED of 6000 cGy [77]. Five out of twelve patients (41%) achieved a response rate with four CRs and one PR, respectively. Three patients (25%) experienced grade 3 side effects. Thus, those preliminary studies suggest that CPIs may be safely combined with radiotherapy. 

## 7. Proposed Algorithm for Older Patients with Locally Advanced Cutaneous Head and Neck Cancer

We propose that older patients (65 years old or above) with unresectable locally advanced head-and-neck cSCC should undergo PD-L1 testing at diagnosis. Their frailty status should be assessed with the G8 screening questionnaire [78]. Even though there are many screening questionnaires which have been validated in clinical trials such as the Omega score, the G8 screening questionnaire is simple to administer in a busy clinic, and is more practical in emerging countries where access to nomogram may be limited [79]. Those with a score of 15 or above will be defined as fit. Those with a score of 14 or less will undergo a complete geriatric assessment with the CGA survey [80]. 

Patients with a positive PD-L1 defined as 1% or higher should undergo immunotherapy with CPIs every three weeks for 12 weeks (four cycles) [36]. Hypofractionated radiotherapy will be started at week 12 concurrently with immunotherapy or earlier for disease progression. Fit patients may be treated to a dose of 5250 cGy in 350 cGy/fraction (BED = 7087 cGy) or a similar BED regimen over four to five weeks depending on tumor location to ensure a good cosmesis. Frail patients may be treated with a weekly hypofractionation of 700 cGy for six weeks (BED = 7140 cGy) per clinician discretion [67]. Immunotherapy will be continued every three weeks following radiotherapy until disease progression, excessive toxicity, or physician discretion to ensure a balance between treatment toxicity and patient QOL for this vulnerable population.

Patients with a negative PD-L1 defined as less than 1% will be started on radiotherapy with a similar regimen. A repeated biopsy will be repeated one week after radiotherapy is completed to assess PD-L1 status. The repeated biopsy is necessary to determine if there is any difference in PD-L1 response among different ethnicities as Asians have been reported to have a better survival following immunotherapy compared to Caucasians [81]. All patients, regardless of PD-L1 status, will be started on immunotherapy four to six weeks after radiotherapy until disease progression, excessive toxicity, or physician discretion. Figure 1 summarizes the IGRG protocol.

Efficacy, toxicity, and impact of the combined treatment on patient QOL would be assessed. The data obtained may guide clinicians to conduct further randomized studies, testing treatment efficacy among different ethnic groups and genders as current clinical trials are biased toward male Caucasians [82,83,84,85,86,87,88]. The undertreatment of women and minorities with head and neck cancer has been reported and may result in a poorer outcome [81,84]. On the other hand, women with head and neck cancer may have a better response to immunotherapy due to possibly a longer half-life of CPIs in women [89,90]. Thus, clinical trials should be stratified to take into consideration differences in age, sex, and ethnicity in survival analysis.

We would like to emphasize the need for prospective studies to validate our hypothesis that combining CPIs and hypofractionated radiotherapy may improve the clinical response rate, thus potentially improving local control and survival for older patients with unresectable locally advanced cSCC. Collaboration among multidisciplinary teams and prospective research efforts will be pivotal in refining treatment strategies for this specific patient population. As an international organization with a large patient network of over 1280 cancer institutions, the IGRG is committed to conduct those studies once we successfully obtain the funding [22,23,24].

## 8. Conclusions

Immunotherapy combined with hypofractionated radiotherapy may be beneficial for older patients with locally advanced unresectable head-and-neck cSCC. Those with negative PD-L1 at diagnosis may benefit from induction hypofractionated radiotherapy to upregulate PD-L1 expression for a potentially improved immune response. Prospective studies should be conducted in the future to verify this hypothesis. 

## Figures and Tables

**Figure 1 cancers-15-04981-f001:**
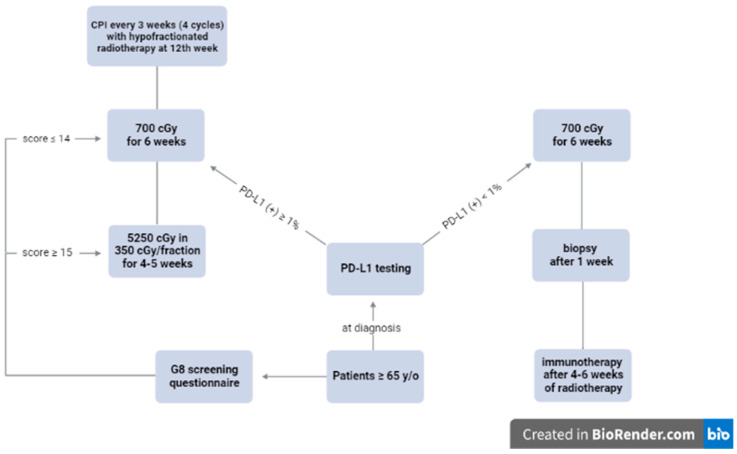
Proposed algorithm by the International Geriatric Radiotherapy Group. The IGRG protocol, created using https://biorender.com (accessed on 28 May 2023).

**Table 1 cancers-15-04981-t001:** Effectiveness of immunotherapy with checkpoint inhibitors in patients with cutaneous squamous-cell carcinoma.

Study (Ref.)	Patients No.	Agents	Response Rate (%)			Grade ≥ 3 Toxicity (%)	Death (%)
			Total	CR	PR		
Gross et al. [36]	79	Cemiplimab	64	51	13	18	5
Migden et al. [37]	78	Cemiplimab	44	13	31	44	1.2
Valentin et al. [38]	22	Cemiplimab	32	9	23	45	4.5
Maubec et al. [39]	57	Pembrolizumab	42	7	35	7	1.3
Hughes et al. [40]	159	Pembrolizumab	40.3	12.6	27.7	11.5	NS
Munhoz et al. [41]	24	Nivolumab	58.3	0	58.3	25	NS
Clingan et al. [42]	103	Cosibelumab	47	9	38	36	NS
Samaran et al. [44]	63	Cemiplimab Pembrolizumab Nivolumab	57.1	19	38.1	28.5	4.7
Strippoli et al. [45]	30	Cemiplimab	76.7	30	46.7	10	NS

No, Number; NS, not stated; ref, reference.

## Data Availability

The data presented in this study is available in this article.

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
