# Peer review of "Immunotherapy and Hypofractionated Radiotherapy in Older Patients with Locally Advanced Cutaneous Squamous-Cell Carcinoma of the Head and Neck: A Proposed Paradigm by the International Geriatric Radiotherapy Group"

_cancers, 2023, doi:10.3390/cancers15204981_

Round 1

Reviewer 1 Report

The manuscript by Nguyen et al is an interesting “perspective” about the convenience of the administration of immunotherapy and hypofractionated radiotherapy in older patients with advance cutaneous SCCs.

Authors provide extensive references reviewing studies relevant for their manuscript. Although there are recent studies reviewing the application of immunotherapy and radiotherapy in aged patients with cSCCs, the perspective of the authors in this manuscript is interesting.

There is an important problem of writing, since there are frequent errors, some of them relevant to understanding the manuscript. Authors should review it carefully. Some examples of errors:

Page 3: “nizolumab”

Page 6: “The feasibility of radiotherapy concurrently with radiotherapy…”

Page 6: Missing or extra words “ We propose that older patients …….. head and neck cSCC head and should undergo…”

Page 6: …”Even though their are many…”

Author Response

We highlight in bold the changes made in the text to address the reviewers concern

Reply to reviewer 1:

Thank you so much for drawing our attention to those errors. We have made corrections in the edited manuscript

Reviewer 2 Report

1. "Surgery" is a discipline, not an event. The correct term for the event is "operation" or "resection" or "surgical resection". 

2. Better justify the use of hypofractionated RT as opposed to other schedules 

3. Your algorithm is based on the idea that the level of PD-1 expression in the tumor is correlated with outcome and it has been made into the critical decision point in your algorithm. What is the data for that? I don't see any presented.  As you may know, the level of PD-1 expression has NOT been shown to be critical in melanoma. 

4. Please correct the subtle errors in English grammar. These are mostly plurality errors. 

Contains subtle errors, mostly in plurality 

Author Response

  1. We replace surgery with surgical resection.
  2. Hypofractionated radiotherapy is very well tolerated and provide good local control in older cancer patients with non-melanoma skin cancer and head and neck cancer in two systemic reviews: page 6, reference 73 and 74. In addition, in the era of COVID-19, a hypofractionation regimen reduces exposure to the virus in this population who are vulnerable to the virus due their age, comorbidity, and immunosuppression from cancer (75).
  3. There are data that support the role of PD-L1 expression in cutaneous squamous cell carcinoma in studies where PD-L1 was included (ref36,46)

Page 3: patients with positive PD-L1 (>1%) had a better clinical response. The pathological complete response was  20% and 40% for patients with PD-L1 expression <1% and 1% or more, respectively. The clinical response was also better for those with positive PD-L1.

Page 5: In another study of 11 patients with recurrent or metastatic cSCC who responded to CPI, a complete response is observed among patients who had a PD-L1 expression of at least 30%.

Those two studies emphasize that we need to include PD-L1 as biomarkers in the skin biopsy specimen to investigate the response of cSCC to immunotherapy.

4, We correct the grammar mistakes

Reviewer 3 Report

This article highlights the significance of developing a new treatment algorithm for the geriatric population suffering from cutaneous squamous cell carcinoma (cSCC), particularly in the head and neck region. The prevalence of cSCC increases with age, primarily due to prolonged exposure to ultraviolet light, making it a disease that predominantly affects older patients. However, the challenge arises when older individuals may not be suitable candidates for surgical interventions due to factors such as frailty and an elevated risk of complications.

Traditionally, surgery coupled with adjuvant radiotherapy has been recommended for locally advanced cSCC to reduce the chances of loco-regional recurrence. Nonetheless, this approach may not always be feasible for older cancer patients. Furthermore, cSCC, especially when poorly differentiated or involving perineural invasion, can be prone to local recurrence and metastasis. Hence, there is a pressing need for a novel treatment algorithm tailored specifically to older patients with locally advanced head and neck cSCC, aiming to enhance their survival rates while maintaining their quality of life.

The article also highlights the potential benefits of immunotherapy with checkpoint inhibitors (CPI) for cSCC patients. This is particularly intriguing due to the high prevalence of program death ligand 1 (PD-L1) in cSCC, which makes it a promising target for this approach. The results from CPI administration have shown a high response rate with acceptable toxicity levels. For those individuals with residual disease, combining hypofractionated radiotherapy with immunotherapy is suggested as a means to reduce the risk of recurrence, as radiotherapy can potentially enhance the effects of immunotherapy. This review is adequate although it suffers from the personal experience of the authors. Have you thought about adding any pilot data?

It is essential to emphasize the need for prospective studies to validate this hypothesis and ensure its efficacy in real-world clinical settings.

The integration of immunotherapy and radiotherapy may represent a promising solution, but further research and clinical studies are necessary to confirm its effectiveness. Collaboration among multidisciplinary teams and prospective research efforts will be pivotal in refining treatment strategies for this specific patient population.

Author Response

We highlight in bold the changes made in the text to address the reviewers concern

  1. We edit the text per recommendation: page 2

2. Unfortunately, we do not have any pilot data. We are applying for grants to conduct them (page 8)

3 We edit the text per reviewer recommendation: page 8